# Associations between Delayed Introduction of Complementary Foods and Childhood Health Consequences in Exclusively Breastfed Children

**DOI:** 10.3390/nu15153410

**Published:** 2023-07-31

**Authors:** Eun Kyo Ha, Seung Won Lee, Ju Hee Kim, Eun Lee, Hye Ryeong Cha, Bo Eun Han, Jeewon Shin, Man Yong Han

**Affiliations:** 1Department of Pediatrics, Hallym University Kangnam Sacred Heart Hospital, Seoul 07441, Republic of Korea; 2School of Medicine, Sungkyunkwan University, Suwon 16419, Republic of Korea; lsw2920@gmail.com (S.W.L.); ryoung0156@gmail.com (H.R.C.); 3Department of Pediatrics, Kyung Hee University Medical Center, Seoul 02447, Republic of Korea; 2004052@gmail.com; 4Department of Pediatrics, Chonnam National University Hospital, Chonnam National University Medical School, Gwangju 61469, Republic of Korea; unelee@daum.net; 5Department of Software, Sejong University, Seoul 05006, Republic of Korea; h.boeunn@gmail.com; 6Department of Pediatrics, Bundang CHA Medical Center, CHA University School of Medicine, Seongnam 13496, Republic of Korea; a186030@chamc.co.kr

**Keywords:** exclusive breastfeeding, complementary feeding, breast milk

## Abstract

The timing of complementary food (CF) introduction is closely related to childhood health, and it may vary depending on the region, culture, feeding type, or health condition. Despite numerous studies on the benefits of breastfeeding and the optimal timing of CF introduction, there have been limited investigations regarding delayed CF introduction in exclusively breastfed children. We compared an exposed group (CF introduction ≥7 months) with a reference group (CF introduction at 4 –< 7 months) regarding hospital admission, disease burden, and growth until age 10. Data from a nationwide population-based cohort study involving children born between 2008 and 2012 in the South Korea were analyzed. The final cohort comprised 206,248 children (165,925 in the exposed group and 40,323 in the reference group). Inverse probability of treatment weighting with propensity score matching was used to balance baseline health characteristics in the comparison groups. We estimated the incident risk ratios (IRR) for outcomes using modified Poisson regression and weighted odds ratios (weighted ORs) and their 95% confidence intervals (CIs) using multinomial logistic regression. The exposed group was associated with low height-for-age *z*-score (HAZ) (IRR (95% CI) for −1.64 < HAZ ≤ −1.03: 1.11 (1.08 to 1.14); HAZ ≤ −1.64: 1.21 (1.14 to 1.27)) and frequent (≥6 events) hospitalizations (weighted OR 1.18 (1.09 to 1.29). The rates of hospital admission, death, and specific medical conditions did not differ between groups. However, delaying the introduction of CF until seven months in exclusively breastfed infants was associated with frequent hospitalization events and lower heights.

## 1. Introduction

The introduction of complementary food (CF) during infancy involves providing food or liquids in addition to breast milk [1]. The timing of CF introduction during infancy is widely acknowledged to have potential effects on health status and growth during later childhood and even adulthood [2]. Previous studies have investigated the optimal transition period from milk to CF [1,3,4,5].

Considerable debate regarding the advantages of breastfeeding and the optimal timing for introducing complementary food (CF) exists [4]. Recently, the optimal time for CF introduction has come under scrutiny, considering the consequent links with the risk of food allergies, excessive weight gain, and growth faltering [5]. Moreover, owing to the differences in infants’ characteristics across countries and variations in awareness of complementary feeding among parents and clinicians, the debate over the optimal time for CF introduction remains unresolved. Previous recommendations may benefit most infants; however, evaluating growth and developmental outcomes within specific populations is necessary for accurate nutrition knowledge and public health [6,7,8,9]. 

We designed this study to examine the effects of delayed CF introduction on the health outcomes of children in the South Korea. Specifically, we compared the morbidity, mortality, burden of specific diseases, and childhood growth up to the age of 10 among exclusively breastfed children. We compared those who had delayed CF to seven months of age with those who were introduced to CF between four and six months.

## 2. Materials and Methods

### 2.1. Study Population and Setting

This nationwide cohort study initially included all individuals born in the South Korea between 1 January 2008, and 31 December 2012 (N = 2,395,966). The observation period was from birth to 31 December 2018, with a mean follow-up duration of 98.4 months. In the South Korea, the National Health Insurance Service (NHIS) provides access to healthcare to over 97% of the population. All data were prospectively recorded, allowing longitudinal surveillance of the entire cohort at the individual level. 

The study population consisted of children whose parent(s) correctly completed the standardized NHIS questionnaire, which included items related to breastfeeding when they were 4 –< 7 and 9 –< 13 months of age. The index date for each child was the date of their first health screening examination at 4 –< 7 months of age. To create a cohort of children that generally reflects healthy children in Korea, children were excluded if they had characteristics that possibly influenced their growth, development, or health such as abnormal birth weight, preterm birth, multiple births, abnormal fetal growth, congenital anomalies, birth trauma, and/or birth hypoxia. Additional exclusion criteria included evidence of a central nervous system disorder or immune disease during the observation period, specific events before the index date (hospitalization for any cause, record of any general anesthesia, admission to the intensive care unit for five or more days), atopic dermatitis, and/or food allergy (as a parent’s concerns regarding allergy might alter feeding practices) [10,11]. Finally, the analysis included data from 467,880 children.

### 2.2. Data Sources and Study Period

Data were obtained from the National Health Screening Program for Infants and Children (NHSPIC) and the Linked NHIS Claims Database. Information on CF introduction time and feeding practices was collected through questionnaires answered by parents [12,13]. Length/height and weight were measured in the screening program when the children were 4 –< 7 months and 9 –< 13 months old and annually until the age of seven. All patient-related records were anonymized, and the study received ethical approval under the current National Health Insurance Act to ensure confidentiality. All recommended guidelines for observational studies using routinely collected health data were followed. The study protocol was reviewed and approved by the Institutional Review Board of the Korea National Institute for Bioethics Policy (number blinded for revision), and the requirement for written consent was waived.

### 2.3. Exposure Measurements

The “exposed group” was defined as children with a delayed introduction to CF until seven months of age, and the “reference group” was introduced to CF between 4 –< 7 months. Specifically, we compared children who did not start CF introduction until the last day of 6 months of age with the control group who started supplementation between 4 –< 7 months. Children were assigned to these groups based on information from the items in the NHSPIC related to CF. At 4 –< 7 months of age, the parents were asked, “What do you usually feed your child?” The possible answers were breastfeeding alone, formula alone, and mixed feeding (breastfeeding and formula). Those who were exclusively breastfed at this age were considered eligible for analysis.

When the child was nine to twelve months, parents were asked, “When did you introduce your child to baby food?” The possible answers were <4 months, 4 –< 7 months, ≥7 months, or not started yet. Based on the responses to these questions, eligible children were classified into the exposed and reference (non-exposed) groups. CF introduction before 4 (<4) months was also defined as an exposure and described elsewhere [2].

### 2.4. Outcome Measurements

Previous studies have shown that breastfeeding was associated with clinical illness and diseases [14,15]. Therefore, the primary outcome of this study investigated all-cause hospitalization which serves as an indicator of health burden by measurable costs. Data were obtained from healthcare databases, and hospital admission was defined as admission to an inpatient facility for a duration of 6 h or more. Repeated hospital admissions were counted as separate episodes when the interval between them was more than one week. The two additional outcomes were intensive care unit (ICU) admission and all-cause mortality. The proportion of hospitalized participants was recorded as the number of events. 

The secondary outcomes focused on child-related health consequences related to breastfeeding, based on a systematic review supporting current World Health Organization (WHO) recommendations [16,17]. These secondary outcomes were used to estimate the real-world effects of the delayed introduction of CF. These clinical conditions were assessed after 24 months, except for the diagnosis of attention-deficit/hyperactivity disorder (ADHD), which was assessed after 48 months, based on previous validation studies [13]. All analysis codes were based on the diagnostic categories from the International Classification of Diseases, 10th Revision (ICD-10), utilizing previously validated definitions (Appendix A). Since the utilization of hospital resources is closely linked to clinical diagnosis and relevant conditions, medical records were used to analyze diagnoses of pneumonia, asthma, upper respiratory tract infection (URTI), acute gastroenteritis, acute otitis media, tooth decay, ADHD, autism spectrum disorder, epilepsy, and malignancy. 

The final anthropometric measures were obtained from the health-screening program when the children were 5–7 years old and were used to analyze growth. Weight percentiles and height-for-age *z*-scores (HAZ) were recorded. Based on the Korean National Child Growth Standards [18], overweight was defined as a body mass index (BMI) corresponding to the 85th percentile or higher, obesity as a BMI corresponding to the 95th percentile or higher. Low heights were categorized based on HAZ scores: −1.64 < HAZ ≤ −1.03 and HAZ scores ≤ −1.64.

### 2.5. Covariates

Seventy covariates that might have potentially influenced outcomes were used for propensity score matching with overlap weighting to reduce potential confounding factors (Appendix A) [19]. The specific covariates included demographic characteristics (such as sex, household income, residence, birth year, and clinical characteristics; Table 1), anthropometric measures and their changes, hospital visits before the index date (Appendix A), data from the physical examinations, child health status, visual/auditory function, and sleeping habits based on parental observations (Appendix A), as well as clinical or medical conditions from the perinatal period to the index date (Appendix A). Baseline demographic information (Table 1), anthropometric measures, and comorbidities were recorded during inpatient and outpatient hospital visits using the NHIS database from birth to the index date (Appendix A). Physical examination findings and questionnaire responses in the index year were obtained from the NHSPIC. 

The analysis was pre-specified using minimal factors for matching. The outcome rates of the minimally matched samples were analyzed similarly to the fully matched samples. Subsequently, the associations between the exposed group and the outcomes were estimated to assess the robustness of the main results. 

### 2.6. Statistical Analysis

Based on previous studies, it was hypothesized that the time of CF introduction was related to baseline covariates [20,21]. Thus, an inverse probability of treatment weighting (IPTW) was used in the propensity score to balance the two groups in terms of their demographic characteristics and baseline health data (Table 1, Appendix A), and stabilized estimations were used to minimize the error caused by an inflated number from the IPTW. The propensity scores (PS) were derived from the predicted probability of the exposed group compared to the unexposed individuals using a multivariable logistic regression model with adjustment for the covariates chosen a priori. Thus, individuals in the exposed group, who share similar characteristics with patients in the unexposed group, possess a higher propensity score and are consequently “up-weighted” when computing the effect of engagement. This approach was performed to effectively eliminate confounding by the analyzed factors. All standardized differences were found to be insignificant after weighting. The weighted incidence risk ratios (IRRs) and 95% confidence intervals (CIs) were obtained using a modified Poisson regression model to determine the relationship between delayed CF introduction and all-cause hospitalization, ICU admission, mortality, certain predefined diseases known to be related to CF introduction time, and child growth (weight and height).

Multinomial logistic regression was used to determine weighted odds ratios (weighted ORs) and their 95% (CIs), using the frequency of hospitalization events and ICU admissions as independent variables. Furthermore, we sought to confirm whether there were differences in specific admission-related diagnoses. Therefore, common diagnoses for hospitalizations were identified, and the proportions within the exposed and control groups were compared. Logistic regression analysis was applied to analyze significant differences in results according to feeding type (exclusive breastfeeding compared with other feeding types, including formula and mixed feeding). 

Furthermore, sensitivity analysis of the association between CF introduction and the primary outcome was performed within a minimally matched cohort, utilizing the Mahalanobis algorithm with a caliper of 0.01 and multivariable regression with 21 covariates (Table 1, Appendix A).

A two-tailed *p*-value less than 0.01 was considered statistically significant, and a *p*-value below 0.001 was used to account for type 1 errors in the presence of multiple comparisons. All analyses were conducted using the SAS Enterprise Guide version 7.1 (SAS Institute Inc., Cary, NC, USA).

## 3. Results

### 3.1. Characteristics of Participating Children

The entire unmatched cohort consisted of 206,248 participants, born between 2008 and 2012, who completed the first two rounds of the health-screening program and were exclusively breastfed for at least four months (Figure 1). Among them, 165,925 infants (80.4%) were introduced to CF between 4 –< 7 months (non-exposed group), and 40,323 (19.6%) at or after 7 months (exposed group). After applying stabilized IPTW, the pseudo-population consisted of 39,055 participants in the exposed group and 160,897 in the non-exposed group (Table 1). Prior to weighting, the standardized mean differences (SMDs) ranged from 0 to 7%, and these differences were substantially reduced after propensity score weighting using the covariates presented in Appendix A. Particularly, after IPTW (N = 199,952), the two groups showed no significant differences in anthropometric measures, hospital visits, or clinical characteristics (Appendix A).

### 3.2. Effect of Delayed CF Introduction on the Risk of Hospitalization and Death 

All-cause hospitalization (the primary outcome) occurred in 53,758 children (33.41%) in the non-exposed group and 13,061 children (33.44%) in the exposed group (Table 2). However, this difference was not statistically significant (weighted IRR,1.00; 95% CI: 0.98–1.02). The study also analyzed the relationship between the duration of CF introduction with all-cause ICU admission, and all-cause death. The risk for ICU admission showed marginal significance (weighted IRR 1.34, 95% CI: 1.01–1.79, *p* = 0.0424), but this difference was not significant after adjusting for multiple comparisons. The two groups showed no differences in terms of the risk of all-cause death (IRR,1.21; 95% CI: 0.72–2.02).

### 3.3. Effect of Delayed CF Introduction on the Risk for Different Clinical Diseases 

We analyzed the different clinical diseases between the two groups (Table 3). The results indicated the exposed group and reference group had no significant differences in pneumonia (12.97% vs. 13.08%; weighted IRR, 0.99; 95% CI: 0.96–1.02), asthma (0.94% vs. 0.88%; weighted IRR, 1.06; 95% CI: 0.95–1.19), URTI (5.44% vs. 5.33%; weighted IRR, 1.02; 95% CI: 0.97–1.07), acute gastroenteritis (6.84% vs. 6.64%; IRR, 1.03; 95% CI: 0.99–1.07), acute otitis media (0.75% vs. 0.79%; IRR, 0.95; 95% CI: 0.84–1.08), or tooth decay (1.21% vs. 1.31%; IRR, 0.93; 95% CI: 0.84–1.03). The analysis showed that very few children had ADHD, autism spectrum disorder, epilepsy, or malignancy, and both groups had similar risks for these conditions.

### 3.4. Effect of Delayed CF Introduction on the Risk for Excessive Weight Gain and Short Height

We examined the associations of delayed CF introduction with overweight, obesity, and low HAZ (−1.64 < HAZ ≤ −1.03 and HAZ ≤ −1.64) in the weighted cohort (Table 4). In this cohort, 15.2% and 15.0% of the exposed and reference groups, respectively, were overweight (IRR, 1.01; 95% CI: 0.99–1.04). Obesity was more common in the reference group (7.1% vs. 6.7%; IRR, 1.05; 95% CI: 1.01–1.09), although the significance was marginal.

Meanwhile, the exposed group also had a significantly higher proportion of children with HAZ for a cutoff −1.64 < HAZ ≤ −1.03 (14.7% vs. 13.3%; IRR, 1.11; 95% CI: 1.08–1.14) and for HAZ ≤ −1.64 (4.4% vs. 3.7%; IRR, 1.21; 95% CI: 1.14–1.27).

### 3.5. Effect of Delayed CF Introduction on the Frequency of Admission to the Hospital and ICU

A multinomial logistic regression analysis was conducted to determine the association between delayed CF introduction and the frequency of all-cause hospitalization and ICU admission (Table 5). The results indicated that children in the exposed group had a significant risk of six or more hospitalizations (2.41% vs. 2.05%; weighted Odds Ratio [OR]:1.18; 95% CI: 1.09–1.29). Although the frequent hospitalization rate was higher in the exposed group, the proportions of the 20 most common diagnoses related to hospitalization were similar between the two groups (Appendix A).

## 4. Discussion

### 4.1. Principal Findings

This study aimed to investigate the association between late CF introduction (≥7 months) and health outcomes in exclusively breastfed children, in comparison to introducing CF between 4 –< 7 months. We compared the effects of different CF introduction periods on childhood health outcomes using representative data from a large-scale population-based nationwide cohort of South Korean children (N = 206,248). There were no significant differences in the all-cause hospitalization, mortality, or disease burden between the two groups during the observation period. However, we found that frequent (≥6) hospitalizations and low HAZ (≤−1.64) were more common in the exposed group. This implies that late CF introduction (≥7 months) in exclusively breastfed infants is associated with more frequent hospital admissions and reduced height. Despite our efforts, we were unable to establish causality between the time of CF introduction and clinical outcomes; conclusive findings can only be obtained through the conduction of prospective studies.

### 4.2. Comparison with Other Studies

There have been studies investigating the timing of weaning food introduction and its potential impact on future health outcomes [6,22,23]. However, there is a scarcity of comparative research on the health outcomes of these children based on the timing of CF introduction in exclusively breastfed children [16]. Additionally, the optimal time to introduce CF in exclusively breastfed children may differ between developing and developed countries. 

Some studies have concluded that prolonged exclusive breastfeeding can reduce the risk of common pediatric infectious diseases, particularly in developing countries. Specifically, infants who were exclusively breastfed for a duration of six months, compared to four months, exhibited a reduced risk of respiratory and gastrointestinal tract infections [24]. Previous studies partially explored the combined effects of breastfeeding and the introduction of CF [15,24,25,26], or focused on early CF introductions [2,27] This highlights the need for research comparing the effects of late CF introductions. Our findings indicated that late CF introduction after 7 months, compared to introducing CF between 4 –< 7 months, did not result in significant childhood diseases. Both groups exhibited no differences in the rates of common respiratory or gastrointestinal infections nor in other major childhood diseases, including asthma, pneumonia, epilepsy, ADHD, autism spectrum disorder, and malignancy [5,28]. 

An important finding of this study was that delayed CF introduction until seven months was associated with low HAZs. The associations were pronounced in children with an infrequent and limited introduction to CF. Although exclusive breastfeeding offers advantages over other forms of milk in terms of childhood growth, our findings suggest that when breastfeeding is not properly combined with CF, it may have adverse consequences for child growth. 

We also examined the effects of delayed CF introduction on hospitalization events, ICU admissions, and mortality, as these can serve as surrogate markers of a child’s health and reflect the severity of diseases, and the associated costs were measurable. Although both groups were similar in all-cause hospitalization and mortality rates, the late CF-introduced group had an increased risk of ICU admission events (≥2 events) and frequent hospitalizations (≥6 events). 

Based on our findings, CF should be introduced before seven months of age, considering possible growth faltering in developed countries, since late CF introduction may result in low HAZs.

### 4.3. Clinical Implications

Although exclusive breastfeeding offers clear benefits, the appropriate introduction of CF is likely to provide better growth and health for children in developed countries where nutritional foods are readily available. Consideration of the optimal duration of CF introduction should take into account the differing contexts between developing and developed countries. 

### 4.4. Strengths and Limitations

One notable strength of this study lies in its investigation of potential outcomes related to delayed CF introduction in breastfed infants using samples derived from a general population-based nationwide cohort, encompassing almost 98% of children born in the South Korea during 2008–2009, in conjunction with national claims data. Since all the data were collected by pediatricians, the likelihood of measurement errors was minimized. Another strength of our study is the utilization of IPTW to ensure that the groups had similar baseline characteristics, thereby minimizing the effects of selection bias and potential confounding factors. Furthermore, we conducted additional analyses to further support our main findings. The strengths of this study provide meaningful “real-world” insights into the timing of CF introduction in exclusively breastfed children. However, this study had several limitations that must be addressed. This study was conducted as an observational study, which means that causality cannot be definitively determined. The children included in our study were all from the South Korea; therefore, the generalizability of the study findings to other developed countries and particularly to developing countries, remains uncertain. Additionally, due to the possibility of misclassification of outcomes, there is a concern about the study being focused solely on confirmed cases recorded as ICD codes in the electronic chart. However, there were no reasons to believe that misclassification differed between the exposed and reference groups. Furthermore, owing to the observational study design, we could not infer causality regarding the relationship between the time of CF introduction and clinical outcomes. Definitive conclusions can only be drawn after replication studies are conducted.

## 5. Conclusions

Our study suggests that late introduction of CF (≥7 months) in exclusively breastfed infants is associated with lower heights during childhood and higher frequency of hospitalizations. However, we did not find any significant associations with high BMI percentiles or clinical issues in childhood, including respiratory diseases (pneumonia, asthma, URTI, acute otitis media), gastroenteric diseases (acute gastroenteritis), tooth decay, neurodevelopmental disorders (ADHD, autism spectrum disorder, epilepsy), or malignancy. These findings can provide valuable insights to clinicians and parents in developed countries, enabling them to make informed decisions regarding the risks and benefits associated with the optimal time of CF introduction for exclusively breastfed children.

## Figures and Tables

**Figure 1 nutrients-15-03410-f001:**
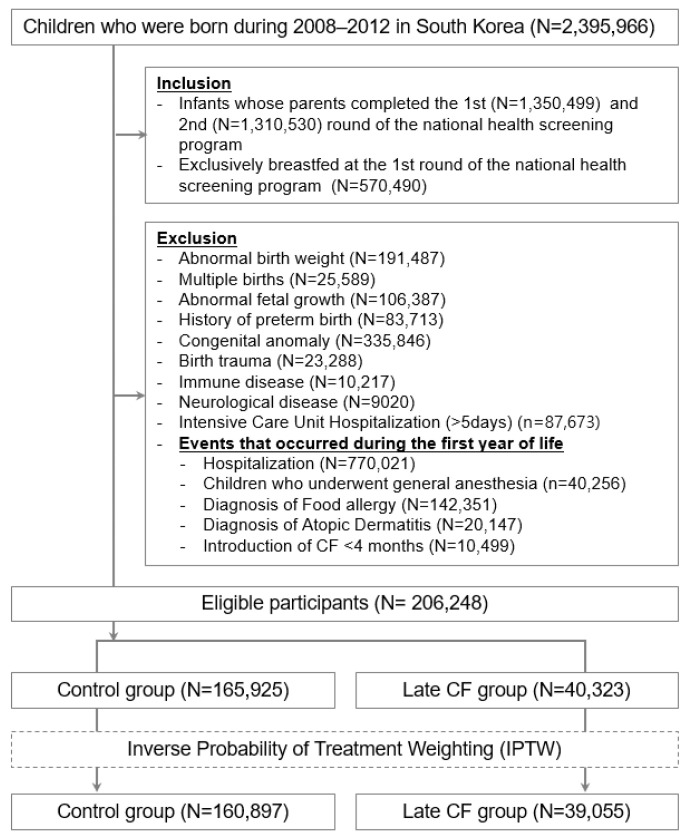
Identification, enrollment, and disposition of participating children. Abbreviations: complementary food, CF; inverse probability of treatment weighting, IPTW.

**Table 1 nutrients-15-03410-t001:** Baseline demographic characteristics of children born between 2008 and 2012 who participated in the screening program ^a, b^.

Demographic Characteristic	Observed Data (N = 206,248)			Weighted Data (N = 199,952) ^c^		
	Non-Exposed (N = 165,925)	Exposed (N= 40,323)	SMD, % ^d^	Non-Exposed (N = 160,897)	Exposed (N = 39,055)	SMD, % ^d^
Sex			1			0
Male	75,738 (45.6)	18,259 (45.3)		73,451 (45.6)	17,837 (45.7)	
Female	90,187 (54.4)	22,064 (54.7)		108,153 (54.4)	21,218 (54.3)	
Birth year						
2008	25,365 (15.3)	5229 (13.0)	7	23,703 (14.7)	5756 (14.7)	0
2009	28,528 (17.2)	7234 (17.9)	2	27,702 (17.2)	6726 (17.2)	0
2010	34,236 (20.6)	8752 (21.7)	3	33,462 (20.8)	8115 (20.8)	0
2011	35,753 (21.6)	8936 (22.2)	2	35,089 (21.8)	8501 (21.8)	0
2012	42,043 (25.3)	10,172 (25.2)	0	40,941 (25.5)	9957 (25.5)	0
Birth residence ^e^						
Seoul	37,868 (22.8)	8097 (20.1)	7	35,895 (22.5)	8712 (22.5)	0
Metropolitan	37,863 (22.8)	9378 (23.3)	1	36,839 (23.1)	8946 (23.1)	0
City	79,493 (47.9)	19,875 (49.3)	3	77,562 (48.6)	18,831 (48.6)	0
Rural	9329 (5.6)	2608 (6.5)	4	9252 (5.8)	2238 (5.8)	0
Income quintile ^f^						
1 (Lowest)	11,246 (6.8)	3038 (7.5)	3	11,075 (7.2)	2694 (7.2)	0
2	20,943 (12.6)	5427 (13.5)	3	20,540 (13.3)	4987 (13.3)	0
3 (Middle)	42,233 (25.5)	10,287 (25.5)	0	40,992 (26.5)	9951 (26.5)	0
4	55,535 (33.5)	12,942 (32.1)	3	53,494 (34.6)	12,983 (34.6)	0
5 (Highest)	29,433 (17.7)	6985 (17.3)	1	28,435 (18.4)	6898 (18.4)	0

Abbreviations: inverse probability of treatment weighting, IPTW; SMD, standardized mean difference. ^a^ The index date of each participant was the date of the first visit for the health screening examination (age 4 –< 7 months). ^b^ Results are reported as n (%) unless otherwise indicated. ^c^ Missing information on the observed data was excluded. IPTW on the propensity score was used to balance the groups for baseline health data. Seventy covariates were used for weighting (Appendix A). Individuals in the non-exposed group were weighed using stabilized weights to create a sample with the same distribution of covariates as the exposed group. ^d^ A difference greater than 10% was considered significant. All standardized differences in the cohort values were insignificant after weighting. ^e^ Metropolitan areas are defined as five metropolitan cities (Busan, Incheon, Gwangju, Daejeon, and Ulsan), urban areas as cities, and rural areas as non-city areas. Missing data in observed data: non exposed group = 1372, exposed group = 365. ^f^ Income was categorized into quintiles of average neighborhood income at the index date. Missing data in observed data: non exposed group = 6535, exposed group = 1644.

**Table 2 nutrients-15-03410-t002:** Weighted relative risks of all-cause hospitalization, ICU admission, and death in the exposed group relative to the reference group ^a, b^.

	Observed Data (N = 206,248)		Weighted Data (N = 199,952) ^c^		
Event ^d^	Non-Exposed (N = 165,925)	Exposed (N = 40,323)	Non-Exposed (N = 160,897)	Exposed (N = 39,055)	Weighted IRR (95% CI) ^e^
All-cause hospitalization	55,268 (33.31)	13,555 (33.62)	53,758 (33.41)	13,061 (33.44)	1.00 (0.98 to 1.02)
All-cause ICU admission	200 (0.12)	66 (0.16)	192 (0.12)	63 (0.16)	1.34 (1.01 to 1.79)
All-cause death	66 (0.04)	21 (0.05)	65 (0.04)	19 (0.05)	1.21 (0.72 to 2.02)

Abbreviations: ICU, intensive care unit; IRR, incidence risk ratio; inverse probability of treatment weighting, IPTW. ^a^ Clinical conditions were recorded after 24 months unless otherwise stated. ^b^ Results are reported as n (%) unless otherwise indicated. ^c^ IPTW on the propensity score was used to balance the groups for baseline health data. Seventy covariates were weighted (Table 1, Appendix A). Individuals in the reference group were weighed using stabilized weights to produce a sample with the same distribution of covariates as the exposed group. ^d^ Events investigated after 24 months. ^e^ Calculated using modified Poisson regression.

**Table 3 nutrients-15-03410-t003:** Weighted risk ratios of clinical conditions in the exposed group relative to the reference group.

	Observed Data (N = 206,248)		Weighted Data (N = 199,952) ^c^		Weighted IRR
Outcomes, N(%) ^a, b^	Non-Exposed (N = 165,925)	Exposed (N = 40,323)	Non-Exposed (N = 160,897)	Exposed (N = 39,055)	(95% CI), % ^d^
Pneumonia	21,628 (13.03)	5274 (13.08)	21,048 (13.08)	5065 (12.97)	0.99 (0.96 to 1.02)
Asthma	1460 (0.88)	381 (0.94)	1416 (0.88)	366 (0.94)	1.06 (0.95 to 1.19)
URTI	8783 (5.29)	2226 (5.51)	8583 (5.33)	2125 (5.44)	1.02 (0.97 to 1.07)
Acute gastroenteritis	10,995 (6.63)	2768 (6.86)	10,688 (6.64)	2672 (6.84)	1.03 (0.99 to 1.07)
Acute otitis media	1314 (0.79)	300 (0.74)	1272 (0.79)	293 (0.75)	0.95 (0.84 to 1.08)
Tooth decay	2181 (1.31)	490 (1.22)	2111 (1.31)	474 (1.21)	0.93 (0.84 to 1.03)
ADHD ^e^	648 (0.39)	142 (0.35)	626 (0.39)	137 (0.35)	0.90 (0.75 to 1.08)
Autism spectrum disorder	276 (0.17)	68 (0.17)	258 (0.16)	62 (0.16)	1.00 (0.76 to 1.32)
Epilepsy	579 (0.35)	130 (0.32)	562 (0.35)	126 (0.32)	0.92 (0.76 to 1.12)
Malignancy	183 (0.11)	49 (0.12)	176 (0.11)	47 (0.12)	1.10 (0.79 to 1.51)

Abbreviations: ADHD, attention-deficit/hyperactivity disorder; IRR, incidence risk ratio; IPTW, inverse probability of treatment weighting; NHIS, National Health Insurance Service; URTI, upper respiratory tract infection. ^a^ Results are reported as n (%) unless otherwise indicated. ^b^ Diseases are listed in Appendix A. To enhance outcome specificity, only the code entered in the main diagnosis field of the database was utilized, as it had the most significant influence on a patient’s hospital stay and resource utilization. Subsequent clinical conditions recorded after 24 months were investigated unless otherwise stated. ^c^ IPTW on the propensity score was used to balance the groups for baseline health data. Seventy covariates were weighted (Table 1 and Appendix A). Individuals in the reference group were weighted using stabilized weights to produce a sample with the same distribution of covariates as the exposed group. ^d^ Calculated using modified Poisson regression. ^e^ Diagnosis after 48 months of age from the NHIS database.

**Table 4 nutrients-15-03410-t004:** Weighted relative risk of increased body mass and height in the exposed group relative to the reference group ^a^.

	Observed Data (N = 206,248)		Weighted Data (N = 199,952) ^c^		Weighted IRR
Outcomes, N(%) ^b^	Non-Exposed (N =165,925)	Exposed (N = 40,323)	Non-Exposed (N = 160,897)	Exposed (N = 39,055)	(95% CI), % ^d^
Overweight ^e^	24 596 (14.82)	6457 (16.01)	24 128 (15.00)	5942 (15.21)	1.01 (0.99 to 1.04)
Obesity ^f^	10 987 (6.62)	3041 (7.54)	10 802 (6.71)	2764 (7.08)	1.05 (1.01 to 1.09)
−1.64 < HAZ ≤ −1.03 ^g^	22 313 (13.45)	5704 (14.15)	21 377 (13.29)	5742 (14.70)	**1.11 (1.08 to 1.14)**
HZA ≤ −1.64 ^g^	6168 (3.72)	1679 (4.16)	5869 (3.65)	1717 (4.40)	**1.21 (1.14 to 1.27)**

Abbreviations: ICU, intensive care unit; IPTW, inverse probability of treatment weighting; HAZ, height-for-age z-score; IRR, incidence risk ratio. Significant differences (*p* < 0.001) are indicated in bold. ^a^ Anthropometric information was recorded at the age of 6–7 years. ^b^ Results are reported as n (%) unless otherwise indicated. ^c^ IPTW on the propensity score was used to balance the groups for baseline health data. Ninety covariates were used for weighting (Table 1 and Appendix A). Individuals in the reference group were weighted using stabilized weights to produce a sample with the same distribution of covariates as the exposed group. ^d^ Calculated using modified Poisson regression. ^e^ Overweight was defined as a BMI measured after five years of age that met or exceeded the 85th percentile of the Korean National Child Growth standards [18]. ^f^ Obesity was defined as a BMI measured after five years of age that met or exceeded the 95th percentile of the Korean National Child Growth standards [18]. ^g^ The age *z*-score range was determined using the Korean National Child Growth standards measure [18].

**Table 5 nutrients-15-03410-t005:** Weighted odds ratios of the number of hospitalizations and ICU admissions in the exposed group relative to the reference group ^a^.

	Observed Data (N = 206,248)		Weighted Data (N = 199,952) ^c^		Weighted OR
All-cause Hospitalization, N (%) ^b^	Non-Exposed (N = 165,925)	Exposed (N = 40,323)	Non-Exposed (N = 160,897)	Exposed (N = 39,055)	(95% CI), % ^d^
None	110,657 (66.69)	26,768 (66.38)	107,139 (66.59)	25,994(66.65)	Reference
1–2 events	40,692 (24.52)	9819 (24.35)	39,565 (24.59)	9488 (24.29)	0.99 (0.96 to 1.02)
3–5 events	11,204 (6.75)	2750 (6.82)	10,899 (6.77)	2632 (6.74)	1.00 (0.95 to 1.04)
≥6 events	3372 (2.03)	986 (2.45)	3294 (2.05)	941 (2.41)	**1.18 (1.09 to 1.29)**
All-cause ICU admission, N (%) ^b^					
None	165,725 (99.88)	40,257 (99.84)	160,705 (99.88)	38,992 (99.8)	Reference
1 event	36 (0.02)	12 (0.03)	34 (0.02)	12 (0.03)	1.39 (0.71 to 2.71)
≥2 events	164 (0.1)	54 (0.13)	158 (0.1)	51 (0.13)	1.34 (0.97 to 1.83)

Abbreviations: CI, confidence intervals; ICU, intensive care unit; IPTW, inverse probability of treatment weighting; OR, odds ratio. A significant difference (*p* < 0.001) is indicated in bold. ^a^ Clinical conditions recorded after 24 months were investigated unless otherwise stated. ^b^ Results are reported as n (%) unless otherwise indicated. ^c^ IPTW on the propensity score was employed to balance the groups for baseline health data. Seventy covariates were weighted (Table 1 and Appendix A). Individuals in the reference group were weighted using stabilized weights to produce a sample with the same distribution of covariates as the exposed group. ^d^ Multinomial logistic regression was used to determine OR and their 95% CI for the number of hospitalization events and ICU admissions as dependent variables. Breastfeeding duration was considered an independent variable.

## Data Availability

This study was based on the National Health Claims Database (NHIS-2020-1-603) established by the NHIS of the Republic of Korea. Applications for using NHIS data are reviewed by the Inquiry Committee of Research Support; if the application is approved, raw data are provided to the applicant for a fee. We cannot provide access to the data, analytic methods, or research materials to other researchers because of the intellectual property rights of this database that is owned by the National Health Insurance Corporation. However, investigators who wish to reproduce our results or replicate the procedure can use the database, which is open for research purposes (https://nhiss.nhis.or.kr/ accessed on 31 June 2022).

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
