# Peer review of "Associations between Delayed Introduction of Complementary Foods and Childhood Health Consequences in Exclusively Breastfed Children"

_nutrients, 2023, doi:10.3390/nu15153410_

Round 1

Reviewer 1 Report

Introduction:

I would like the authors to clarify further why they have chosen the reference group to be inclusive of 6 months (i.e. why is this not an analysis that makes the reference group 4 to <6 months and uses 4 to <7 months. This is very unusual and no clarification or enough evidence in the introduction was really provided as to why that is a new cutoff . This is actually my biggest concern for this paper and strongly recommend that the analysis be changed so that the reference group is 4 to < 6 months

Line 76: In the sentence “Finally, the analysis included data from 467 880 children” perhaps you meant excluded? Please clarify and also note that this number is not reflected in Figure 1. 

Line 98: Can you be a little more specific about the age ranges. My recommendation is to do the analysis with 4-<6 months as the reference group. Can you also categorize infants as those that initiate complementary feeding at 4-<6 months and then those that initiate complementary feeding at 7 months as the exposed group. The recommendation is usually to begin CF 6 months rather than 7 months.

Line 103: This sentence should be re-written to give the direction of the association as it is very confusing the way it is written here. Is hospital admission associated with EBF or is breastfeeding negatively impacted by hospitalization. Also in the next few sentences (lines 104-106) it sounds like utilization of healthcare resources is being equated to all-cause hospitalization. This is unclear. Utilization of healthcare resources can be taken to mean many things (e.g. lactation consultation) so it is very unclear the way it is written and hard to equate this to all-cause hospitalization

Line 104-106: This is unclear, the utilization of healthcare resources is not quite equivalent to all-cause hospitalization. I recommend being very explicit about this.

Line 126: The HFA is not a commonly used term. The term  should be changed to height-for-age z-scores and is abbreviated as HAZ rather than HFA z-score.

Table 1: 

Tabel 1 needs to be re-checked and missing information should be added as a separate row or the number of children not included due to missing should be added as a footnote with explanation. please include after footnote c whether the change in sample size is due to missing information on covariates and the number of children excluded from the weighted data. 

This is minor but the sex percent should add to 100% (careful with rounding for males)

Looking at the income quintile:  for the non-exposed group should there be a row with missing n (%) as the N’s do not add up to 165 925 nor does the percent add up to 100%. Also similar to the exposed group.If there are children with missing information then this should be noted on a separate row. 

I understand that the number of children will fluctuate from the observed to the weighted data however it is unclear how it would be possible that the number of children in the observed data and the weighted data increases when we look at the same variable in the weighted data. For example, for children in the 4th income quintile, there are 12 942 children that are exposed (observed data) and 12 983 exposed (weighted data) or for birth place the exposed children born in Seoul was 8097 in the observed data and this increased to 8712 in the weighted data. I also see a similar trend for males l. 

Line 192: since you have introduced the height-for-age z-score abbreviation you can use it throughout the manuscript

Figure 1: 

In the exposed group, the figure shows that the exposure is defined as CF is delayed until 6 months of age. Do you mean until 7 months of age or after 6 months of age instead? It would be helpful to have very specific age ranges (e.g. 4-<6 months, or 4 - <7 months, or 4 -6.9 months as an example).

Line 273/Table 4: 

In this analysis you are trying to look at poor linear growth so you can either use height-for-age and then look at the percentiles (<15% or <5%) or you can look at the height-for-age z scores <-2 (for example). I have never seen a HAZ cutoff with percentiles and the way it is presented here appears to be conflating the two. Please change this to either one. I have a similar comment for the BMIZ. The BMI z-score is a measure of how many standard deviations a child's BMI is above or below the average BMI for their age and gender. You can take  percentiles of BMI (before converting it to z scores)

Discussion:

I would like the concluding statement to note that this is observational and so causality cannot be made. Also, if the analysis is to stay between 4-6 months as the referent group 7 months for the exposure group then I would like to see additional studies that use the same cutoffs

Line 349-351:This sentence notes that studies that promotes CF for improved health outcomes  are less reliable because they are observational studies. What makes the findings from this study different? Also, there are far more studies about CF then the two that are provided and I would like to see additional comparison.

Line 394: Here, I recommend changing this since it is contradictory to the later limitation you highlight on line 402. Perhaps here you mean that it is generalizable to the broader pediatric population within South Korea?

Line 406: add “of outcome” after misclassification. 

Line 407:Do you think there was a misclassification of the exposure since this is based on parental recall? How would this impact the results 

Line 407- 411: I think this is an extremely important point and should be moved to the beginning of the discussion section rather than as a last row of the limitations section. This should be moved to be included in paragraph between 333-345.

Line 411-413: The conclusion statement needs to be clarified. It is not clear what you mean by ‘without significant risk for hospital resources’ or what you mean by ‘specific disease rates’.

Author Response

Response to Reviewer 1 Comments

Introduction:

I would like the authors to clarify further why they have chosen the reference group to be inclusive of 6 months (i.e. why is this not an analysis that makes the reference group 4 to <6 months and uses 4 to <7 months. This is very unusual and no clarification or enough evidence in the introduction was really provided as to why that is a new cutoff . This is actually my biggest concern for this paper and strongly recommend that the analysis be changed so that the reference group is 4 to < 6 months

Response) Thank you for your insightful comments. As you mentioned, authoritative nutrition societies, including WHO (https://www.who.int/health-topics/complementary-feeding#tab=tab_1), recommend initiating weaning at 6 months. However, the authors hypothesized in this study that inappropriately delayed weaning in breastfed infants could have adverse effects on growth and weight gain. Previous literature presenting that the dietary pattern changed in this period was also referenced in dividing the period (Medicinia 2018). For this reason, the exposed group was defined as those who commenced weaning after 7 months, which is considered too late. We further explained this in the method section to address any potential confusion arising from this exposure. Your feedback is greatly appreciated. Thank you once again.

Methods) Specifically, we compared children who did not start CF introduction until the last day of 6 months of age with the control group who started supplementation between 4–<7 months.

Line 76: In the sentence “Finally, the analysis included data from 467 880 children” perhaps you meant excluded? Please clarify and also note that this number is not reflected in Figure 1. 

Response) Thank you for your important comment. I've acknowledged the part you mentioned was incorrectly described, and I've taken the necessary steps to correct it.

Finally, 165,925 and 40,323 children were included in the control and late CF groups, respectively.

Additionally, we made adjustments to the picture to enhance its clarity.

Line 98: Can you be a little more specific about the age ranges. My recommendation is to do the analysis with 4-<6 months as the reference group. Can you also categorize infants as those that initiate complementary feeding at 4-<6 months and then those that initiate complementary feeding at 7 months as the exposed group. The recommendation is usually to begin CF 6 months rather than 7 months.

  1. R) Thank you for your insightful comments. As you mentioned, authoritative nutrition societies, including WHO (https://www.who.int/health-topics/complementary-feeding#tab=tab_1), recommend initiating weaning at 6 months. However, the authors hypothesized in this study that inappropriately delayed weaning in breastfed infants could have adverse effects on growth and weight gain. Previous literature presenting that the dietary pattern changed in this period was also referenced in dividing the period (Medicinia 2018). For this reason, the exposed group was defined as those who commenced weaning after 7 months, which is considered too late. We further explained this in the method section to address any potential confusion arising from this exposure. Your feedback is greatly appreciated. Thank you once again.

Methods) Specifically, we compared children who did not start CF introduction until the last day of 6 months of age with the control group who started supplementation between 4–<7 months.

Line 104-106: This is unclear, the utilization of healthcare resources is not quite equivalent to all-cause hospitalization. I recommend being very explicit about this.

  1. R) We agree that utilization of healthcare resources is not quite equivalent to all-cause hospitalization. We revised the part as follows:

 Previous studies have shown that breastfeeding was associated with clinical illness and diseases [14, 15]. Therefore, the primary outcome of this study investigated all-cause hospitalization which serves as an indicator of health burden by measurable costs.

Line 126: The HFA is not a commonly used term. The term should be changed to height-for-age z-scores and is abbreviated as HAZ rather than HFA z-score.

 Thank you for bringing up this valuable point. As you've mentioned, I agree that "HZA" seems to be a more suitable abbreviation than "HFA." Therefore, the text has been updated by replacing the previously referred "HFA" with "HZA" instead. Your input is greatly appreciated.

Table 1: 

Tabel 1 needs to be re-checked and missing information should be added as a separate row or the number of children not included due to missing should be added as a footnote with explanation. Please include after footnote c whether the change in sample size is due to missing information on covariates and the number of children excluded from the weighted data. 

This is minor but the sex percent should add to 100% (careful with rounding for males)

 Looking at the income quintile:  for the non-exposed group should there be a row with missing n (%) as the N’s do not add up to 165 925 nor does the percent add up to 100%. Also similar to the exposed group .If there are children with missing information then this should be noted on a separate row. 

 Thank you for the important comments.

We carefully reviewed the numbers of Table 1 and corrected the errors. We also added information regarding the missing numbers.

c Missing information on the observed data was excluded. IPTW on the propensity score was used to balance the groups for baseline health data. Seventy covariates were used for weighting (Supplementary Tables 3, 4). Individuals in the non-exposed group were weighed using stabilized weights to create a sample with the same distribution of covariates as the exposed group.
d A difference greater than 10 % was considered significant. All standardized differences in the cohort values were insignificant after weighting.
e Metropolitan areas are defined as five metropolitan cities (Busan, Incheon, Gwangju, Daejeon, and Ulsan), urban areas as cities, and rural areas as non-city areas. Missing data in observed data; Non exposed group =1372, exposed group =365

f Income was categorized into quintiles of average neighborhood income at the index date. Missing data in observed data; Non exposed group =6535, exposed group = 1644

I understand that the number of children will fluctuate from the observed to the weighted data however it is unclear how it would be possible that the number of children in the observed data and the weighted data increases when we look at the same variable in the weighted data. For example, for children in the 4th income quintile, there are 12 942 children that are exposed (observed data) and 12 983 exposed (weighted data) or for birth place the exposed children born in Seoul was 8097 in the observed data and this increased to 8712 in the weighted data. I also see a similar trend for males l. 

Thank you for your important comment regarding the statistical analysis. We have revised the manuscript to provide more detailed explanations of the models used in our study. 

The Propensity scores (PS) were derived from the predicted probability of the exposed group compared to the unexposed individuals using a multivariable logistic regression model with adjustment for the covariates chosen a priori. Thus, individuals in the exposed group, who share similar characteristics with patients in the unexposed group, possess a higher propensity score and are consequently "up-weighted" when computing the effect of engagement. This approach was performed to effectively eliminate confounding by the analyzed factors.

Specifically, we employed the stabilized inverse probability of treatment weighting (IPTW) method with propensity score (PS) matching to compare the exposed and unexposed (control) groups. PS was calculated using a multivariate logistic regression model with pre-selected covariates. The control and CH groups were then weighted using the formula of unexposed proportion/(1-PS) and exposed proportion/PS, respectively. Then, we estimated the average treatment effect, which produced a weighted pseudo sample of participants in the reference group with the same distribution of measured covariates as the exposed group. After weighting, participants were allocated to the unexposed (control) groups and exposed groups, respectively. There were cases where the number in weighted data increased after going through these formulas, as in the case mentioned

We have provided a more detailed description of the statistical analyses in the methods section as follows:

Line 192: since you have introduced the height-for-age z-score abbreviation you can use it throughout the manuscript

  1. R) Thank you. We introduced the height-for-age z-score abbreviation and used it throughout the manuscript.

Figure 1: 

In the exposed group, the figure shows that the exposure is defined as CF is delayed until 6 months of age. Do you mean until 7 months of age or after 6 months of age instead? It would be helpful to have very specific age ranges (e.g. 4-<6 months, or 4 - <7 months, or 4 -6.9 months as an example).

 Thank you for your feedback regarding Figure 1. Based on your comment, we revise the figure to be more comprehensible. Additionally, we have further refined the description of specific age ranges in the text to provide clarity as follows:

2.3. Exposure measurements

Specifically, we compared children who did not start CF introduction until the last day of 6 months of age with the control group who started supplementation between 4–<7 months.

Line 273/Table 4: 

In this analysis you are trying to look at poor linear growth so you can either use height-for-age and then look at the percentiles (<15% or <5%) or you can look at the height-for-age z scores <-2 (for example). I have never seen a HAZ cutoff with percentiles and the way it is presented here appears to be conflating the two. Please change this to either one. I have a similar comment for the BMIZ. The BMI z-score is a measure of how many standard deviations a child's BMI is above or below the average BMI for their age and gender. You can take percentiles of BMI (before converting it to z scores)

  1. R) Thank you for providing advice regarding description of anthropometric measures. As you mentioned, there was some confusion in the past concerning the usage of percentiles and z-scores. We have carefully considered your comments and conducted additional literature review to ensure the accurate and appropriate expressions are used. We corrected the statements and Table contents as follows:

Weight percentiles and height-for-age z-scores (HAZ) were recorded. Based on the Korean National Child Growth Standards [18], overweight was defined as a body mass index (BMI) corresponding to the 85th percentile or higher, obesity as a BMI corresponding to the 95th percentile or higher. Low heights were categorized based on HAZ scores: -1.64 < HAZ ≤ -1.03 and HAZ scores ≤ -1.64.

The reference group also showed a significantly higher proportion of children with HAZ scores in the range of -1.64 < HAZ ≤ -1.03, and also for HAZ scores ≤ -1.64.

 Table 4. Weighted relative risk of increased body mass and height in the exposed group relative to the reference group.a.

Observed Data (N = 206 248)

Weighted Data (N =199 952) c

Outcomes,

N(%)b

Non-exposed

(n =165 925)

Exposed

(n = 40 323)

Non-exposed

(n = 160 897)

Exposed

(n = 39 055)

Weighted IRR

(95 % CI), %d

Overweight e

24 596 (14.82)

6457 (16.01)

24 128 (15.00)

5942 (15.21)

1.01 (0.99 to 1.04)

Obesity f

10 987 (6.62)

3041 (7.54)

10 802 (6.71)

2764 (7.08)

1.05 (1.01 to 1.09)

-1.64<HAZ ≤-1.03 g

22 313 (13.45)

5704 (14.15)

21 377 (13.29)

5742 (14.70)

1.11 (1.08 to 1.14)

HAZ ≤ -1.64 g

6168 (3.72)

1679 (4.16)

5869 (3.65)

1717 (4.40)

1.21 (1.14 to 1.27)

Abbreviations: ICU, intensive care unit; Inverse probability of treatment weighting, IPTW; HZA, height-for-age z-score; IRR, Incidence Risk Ratio.

Significant differences (P < 0.001) are indicated in bold.
a Anthropometric information was recorded at the age of 6–7 years.

b Results are reported as n (%) unless otherwise indicated.
c IPTW on the propensity score was used to balance the groups for baseline health data. Ninety covariates were used for weighting (Table 1 and Supplementary Tables 3 and 4). Individuals in the reference group were weighted using stabilized weights to produce a sample with the same distribution of covariates as the exposed group.
d Calculated using modified Poisson regression.
e Overweight was defined as a BMI z-score measured after five years of age that met or exceeded the 85th percentile of the Korean National Child Growth standards16.
f Obesity was defined as a BMI z-score measured after five years of age that met or exceeded the 95th percentile of the Korean National Child Growth standards16.
g The age z-score range was determined using the Korean National Child Growth standards measure. Short16

Discussion:

I would like the concluding statement to note that this is observational and so causality cannot be made. Also, if the analysis is to stay between 4-6 months as the referent group 7 months for the exposure group then I would like to see additional studies that use the same cutoffs

  1. R) Thank you for your valuable comment. As this is an observational study, establishing causality was not possible. Recognizing this important limitation, we have now included a section in the text that highlights this aspect.

 This study was conducted as an observational study, which means that causality cannot be definitively determined.

Also, we changed 4-6months to 4-<7months to improved clarity. Although, there were not many studies using this cutoffs (Medicinia 2018)(JPGN 2008). Thank you for the important comment.

Line 349-351: This sentence notes that studies that promotes CF for improved health outcomes are less reliable because they are observational studies. What makes the findings from this study different? Also, there are far more studies about CF then the two that are provided and I would like to see additional comparison.

  1. R) Thank you for your valuable comment. As you rightly pointed out, numerous valuable papers, including systematic reviews, have been published on the ideal timing for introducing CF. However, it is important to note that this study did not focus on determining the ideal timing for introducing CF. Consequently, some of our previous discussions might have been inaccurately described, which could lead to misunderstandings. To address this concern and align with the specific purpose of this study, we have made amendments to the relevant section as follows. We are grateful for your important feedback.

 There have been numerous studies investigating the duration of exclusive breastfeeding, including systematic reviews [6, 22]. However, most of these studies have primarily concentrated on the overall breastfeeding period, and there is a scarcity of comparative research on the health outcomes of these children based on the timing of CF introduction[16, 23].

Line 394: Here, I recommend changing this since it is contradictory to the later limitation you highlight on line 402. Perhaps here you mean that it is generalizable to the broader pediatric population within South Korea?

  1. R) Thank you for your important comment. We agree that this part of the strength is contradictory to the later limitation. We revised the description as follows:

One notable strength of this study lies in its investigation of potential outcomes related to delayed CF introduction in breastfed infants using samples derived from the general population-based nationwide cohort, encompassing almost 98% of children born in South Korea during 2008–2009, in conjunction with national claims data.

Line 406: add “of outcome” after misclassification. 

  1. R) Thank you for the comment, and we revised the text based on your recommendation.

Additionally, due to the possibility of misclassification of outcome, there is a concern since the study focused solely on confirmed cases recorded as ICD codes in the electronic chart.

Line 407: Do you think there was a misclassification of the exposure since this is based on parental recall? How would this impact the results? 

  1. R) Due to the study subjects being infants, the research relied on questionnaires answered by guardians. Consequently, the potential for recall bias exists, as is common in such studies. However, since the screening program involved hospital visits at specific months of age and the information on feeding practices and CF introduction at the time was collected, we believe that impact on the results was minimal.

Line 407- 411: I think this is an extremely important point and should be moved to the beginning of the discussion section rather than as a last row of the limitations section. This should be moved to be included in paragraph between 333-345.

  1. R) The content you mentioned has been adequately addressed in the beginning of the discussion section.

Despite our efforts, we were unable to establish causality between the time of CF introduction and clinical outcomes; conclusive findings can only be obtained through the conduction of prospective studies.

Line 411-413: The conclusion statement needs to be clarified. It is not clear what you mean by ‘without significant risk for hospital resources’ or what you mean by ‘specific disease rates’.

  1. R) Thank you for the important comment. The part you mentioned seems unclear, so we revised the sentences as follows:

Our study suggests that late introduction of CF (≥7 months) in exclusively breastfed infants is associated with lower heights during childhood and more frequent hospitalizations, but not with high BMI percentiles or specific disease rates.

Reviewer 2 Report

Thank you for the opportunity to review this manuscript that examines the association between delayed introduction of complementary foods and longer-term outcomes. The large national dataset used for this study is a huge strength, as are the sophisticated methods used to address potentially confounding factors. Overall, this is well written and well organized, the methods are strong, and the conclusions appropriate. This study addresses a question about timing of CF that has not been adequately addressed in the literature.

Overall comments:

1.     Were there no children who received CF early? This seems unlikely, but these children were not addressed. Please address this in the flow diagram and in the inclusion/exclusion criteria. If they were excluded, why?

2.     How many were excluded because they were not EBF until at least 4 mo? Please include in methods and Figure 1. And describe why they were excluded.

3.     Consistency in how the “exposed group” and “non-exposed group” are reported or described would be helpful. For example is the group defined by EBF duration or by CF introduction? Figure 1 uses EBF>6mo instead of EBF>7 months or even CF began at 4-6 mo and CF began at 7 mo (which is how this is described in the text).

Abstract:

4.     Page 1, line 17-19: Please make it clear in the first couple of sentences that you are examining delayed introduction of CF. In my region, I would expect substantially higher early introduction,  so this is a regional or cultural difference, that just needs to be clear in the abstract. Note that it is clear in the main text.

5.     Page 1, line 27: redundancy with “low height percentiles for height for age.” Remove the first “height”

Methods:

6.     Line 81: Infants are measured by length, not height, so I believe this should say “Length and weight were measured in the screening…”

7.     Line 129-130: It looks like you are using percentiles instead of z-scores to determine low height for age. If this is the case, then I think you can leave z-scores out of it. Z-scores corresponding to a certain percentile is simply the percentile.

8.     Please provide information about how number of CF were collected/assessed. This is mentioned in statistical analysis, but not in exposure measurements, so it is unclear what this is referring to. Also, what was considered “insufficient” (line 329) and what was this based on?

Results:

9.     Line 208:

o   Typo. It should say “…delayed CF introduction until seven months of age (exposed group.” Currently it is says six.

o   Were there no infants who received CF before 4 months of age? If not, this deserves comment. If yes, it seems they were excluded from the sample? But I don’t see this discussed in the results.

10.  Figure 1 flow diagram: where are those who were not EBF? Those were excluded, were they not?

11.  Line 225-226: it seems a word is missing “The risk for ICU admission was higher in the exposed group…” 

12.  Line 278: Here it states that the reference group had higher proportions of low HFA. This is inconsistent with the data in table 4, and presumably a typo?

13.  Lines 327-330:

o   The subheader is not very informative “CF practices” could be referring to many things, such as responsive feeding or quality of CF provided. But here I think this strictly means quantity or number?

o   What is insufficient CF?

o   Please define this in the methods and provide some background info/evidence for defining this.

o   Consider moving these data into the main tables rather than in supplementary files.

o   Overall, I want more information here, particularly since these were significant findings that are then referred to in the discussion (Line 342).

Discussion

14.  Line 334: Presumably another typo calling six months delayed CF introduction when this should say seven.

15.  Lines 360-361: This states that there were “no significant clinical consequences.” I would argue that low HFA and higher hospitalizations are clinical consequences. Suggest modifying this statement.

16.  Lines 374-376: Please revise this sentence. Either a higher number had ICU admission or the group had an increased risk.

17.  Line 379: Based on your findings it would be more accurate to say “…CF should be introduced before seven months of age…” not “no later than.” The latter suggests that 7 months is not delayed, and this is inconsistent with your findings.

18.  Line 386: replace the word duration for timing.

Conclusions

19.   Line 413: “…significant risks for hospital resources…” should be “significant risks for  hospitalization”

Author Response

Response to Reviewer 2 Comments

Thank you for the opportunity to review this manuscript that examines the association between delayed introduction of complementary foods and longer-term outcomes. The large national dataset used for this study is a huge strength, as are the sophisticated methods used to address potentially confounding factors. Overall, this is well written and well organized, the methods are strong, and the conclusions appropriate. This study addresses a question about timing of CF that has not been adequately addressed in the literature.

  1. R) Your valuable review has significantly improved our manuscript. Thank you for your constructive feedback and support. We have made every effort to thoroughly address your concerns. If there are any further points or aspects that require attention, please do not hesitate to let us know.

Overall comments:

  1. Were there no children who received CF early? This seems unlikely, but these children were not addressed. Please address this in the flow diagram and in the inclusion/exclusion criteria. If they were excluded, why?
  2. R) Thank you for raising this crucial point. You mentioned the early introduction of baby food, and we truly appreciate your insight. In this analysis, we have excluded children who were introduced to weaning at an early age; however, we understand that the initial explanation provided may have been insufficient. To address this concern, we have now added the following explanation to both Figure 1 and the method section.

The possible answers were <4 months, 4-<7months, ≥ 7 months, or not started yet. Based on the responses to these questions, eligible children were classified into the exposed group and the reference (non-exposed) group. CF introduction before 4 (<4) months was also defined as an exposure and described elsewhere [2].

  1. How many were excluded because they were not EBF until at least 4 mo? Please include in methods and Figure 1. And describe why they were excluded.

Thank you for the important comment. Out of the 1,350,499 participants who adequately completed the first round of the national health screening, 780,009 reported other forms of feeding, including mixed or formula feeding, or other forms of feeding types and were excluded. We included this information in the method section for clarification. You extend our gratitude for your contribution.

  1. Consistency in how the “exposed group” and “non-exposed group” are reported or described would be helpful. For example is the group defined by EBF duration or by CF introduction? Figure 1 uses EBF>6mo instead of EBF>7 months or even CF began at 4-6 mo and CF began at 7 mo (which is how this is described in the text).
  2. R) Thank you for your insightful comment. I agree that the previous expression, "4-6 months vs ≥7 months," might have caused some confusion. Following your suggestion, I have made the necessary adjustment in the text, now using "4-<7 months vs ≥7 months," to improve clarity.

Results) Among them, 165 925 infants (80.4 %) were introduced to CF between 4-<7 months (non-exposed group), and 40 323 (19.6 %) at ≥ 7 months (exposed group)

Abstract:

  1. Page 1, line 17-19: Please make it clear in the first couple of sentences that you are examining delayed introduction of CF. In my region, I would expect substantially higher early introduction, so this is a regional or cultural difference, that just needs to be clear in the abstract. Note that it is clear in the main text.
  2. R) Thank you for the important comment. I revised the abstract as follows in order to be clear as follows:

We compared an exposed group (CF introduction ≥ 7 months) with a reference group (CF introduction at 4–<7 months) regarding hospital admission, disease burden, and growth until age 10.

  1. Page 1, line 27: redundancy with “low height percentiles for height for age.” Remove the first “height”

 Response) Thank you for the important comment. We revised the part as follows :

The exposed group was associated with low for height for age z-score (HAZ) (IRR (95 % CI) for -1.64<HAZ≤-1.03: 1.11 (1.08 to 1.14); HAZ≤-1.64: 1.21 (1.14 to 1.27)) and frequent (≥6 events) hospitalizations (weighted OR 1.18 (1.09 to 1.29).

Methods:

  1. Line 81: Infants are measured by length, not height, so I believe this should say “Length and weight were measured in the screening…”

 Response) Thank you for the important comment. We revised the part as follows :

Length/height and weight were measured in the screening program when the children were 4–<7 months and 9–<13 months old and annually until the age of seven.

  1. Line 129-130: It looks like you are using percentiles instead of z-scores to determine low height for age. If this is the case, then I think you can leave z-scores out of it. Z-scores corresponding to a certain percentile is simply the percentile.
  2. R) Thank you for raising this important point. I agree that the part regarding z-score and percentile might have been unclear. Consequently, I have made further adjustments to provide a clearer explanation, as follows.

Weight percentiles and height-for-age z-scores (HAZ) were recorded. Based on the Korean National Child Growth Standards [18], overweight was defined as a body mass index (BMI) corresponding to the 85th percentile or higher, obesity as a BMI corresponding to the 95th percentile or higher. Low heights were categorized based on HAZ scores: -1.64 < HAZ ≤ -1.03 and HAZ scores ≤ -1.64.

  1. Please provide information about how number of CF were collected/assessed. This is mentioned in statistical analysis, but not in exposure measurements, so it is unclear what this is referring to. Also, what was considered “insufficient” (line 329) and what was this based on?

Supplementary Table 6 was indeed a post-hoc analysis aimed at highlighting the increased impact on hospitalization or height when the type of weaning food introduced is less diverse. However, upon careful consideration, I realized that this additional analysis might not contribute significantly to the overall findings. Therefore, after much deliberation, I made the decision to remove that part from the study. Your thoughtful comment and feedback have been essential in refining the content.

Removed contents

Methods)

Feeding patterns such as time of solid food introduction, number of solid foods consumed per day, and number of types of food introduced, were considered as independent variables, and aforementioned meaningful outcomes (frequent [≥6 events] and low [<5] hospitalizations and low HAZ) were set as dependent variables for this analysis.

Results)

3.6. Association between CF practices with frequent hospitalizations and low heights

Post hoc analysis revealed significant differences between the two groups in the daily CF consumed. Children with insufficient CF were more likely to experience six or more hospitalizations and have a low HAZs (Supplementary Table 6).

Discussion

Further analysis suggested that these differences were associated with the number of CF consumed per day.

Post hoc analysis revealed that this was related to the insufficient number of CF consumed daily.

Table S6. Post hoc analysis for frequent hospitalizationa, low height percentiles and nutritional practices within children exclusively breastfed ≥7 months.

Results:

  1. Line 208:

o   Typo. It should say “…delayed CF introduction until seven months of age (exposed group.” Currently it is says six.

  1. R) Thank you for the comment. We revised the mentioned statement as follows:

Among them, 165 925 infants (80.4 %) were introduced to CF between 4-<7 months (non-exposed group), and 40 323 (19.6 %) after 7 months (exposed group).

o   Were there no infants who received CF before 4 months of age? If not, this deserves comment. If yes, it seems they were excluded from the sample? But I don’t see this discussed in the results.

  1. R) We agree that children who received CF before 4 months of age deserves a comment. In fact, we separately analyzed these children. We mentioned this in the methods, and in the discussion as well.

Previous studies partially explored the combined effects of breastfeeding and the introduction of CF [15, 25-27], or focused on early CF introductions [2, 28] This highlights the need for research comparing the effects of late CF introductions

  1. Figure 1 flow diagram: where are those who were not EBF? Those were excluded, were they not?
  2. R) Thank you for the important comment. Out of the 1,350,499 participants who adequately completed the first round of the national health screening, 780,009 reported other forms of feeding, including mixed or formula feeding, or other forms of feeding types and were excluded. We included this information in the method section for clarification. You extend our gratitude for your contribution.

  1. Line 225-226: it seems a word is missing “The risk for ICU admission was higher in the exposed group…” 
  2. R) Thank you for the important comment. We corrected the sentence as follows:

The risk for ICU admission showed marginal significance (weighted IRR 1.34, 95% CI: 1.01–1.79, P = 0.0424), but this difference

  1. Line 278: Here it states that the reference group had higher proportions of low HFA. This is inconsistent with the data in table 4, and presumably a typo?
  2. R) Thank you for the important comment. We corrected the sentence and some contents of Table 4 as follows:

Meanwhile, the exposed group also had a significantly higher proportion of children with HAZ for a cutoff -1.64<HAZ≤-1.03 (14.7 % vs. 13.3 %; IRR, 1.11; 95 % CI:1.08–1.14) and for HAZ ≤ -1.64 (4.4 % vs. 3.7 %; IRR, 1.21; 95 % CI:1.14–1.27).

Table 4. Weighted relative risk of increased body mass and height in the exposed group relative to the reference group.a.

Observed Data (N = 206 248)

Weighted Data (N =199 952) c

Weighted IRR

Outcomes, N(%)b

Non-exposed

(n =165 925)

Exposed

(n = 40 323)

Non-exposed

(n = 160 897)

Exposed

(n = 39 055)

(95 % CI), %d

Overweight e

24 596 (14.82)

6457 (16.01)

24 128 (15.00)

5942 (15.21)

1.01 (0.99 to 1.04)

Obesity f

10 987 (6.62)

3041 (7.54)

10 802 (6.71)

2764 (7.08)

1.05 (1.01 to 1.09)

-1.64<HAZ ≤-1.03 g

22 313 (13.45)

5704 (14.15)

21 377 (13.29)

5742 (14.70)

1.11 (1.08 to 1.14)

HZA ≤ -1.64 g

6168 (3.72)

1679 (4.16)

5869 (3.65)

1717 (4.40)

1.21 (1.14 to 1.27)

  1. Lines 327-330:

o   The subheader is not very informative “CF practices” could be referring to many things, such as responsive feeding or quality of CF provided. But here I think this strictly means quantity or number?

o   What is insufficient CF?

o   Please define this in the methods and provide some background info/evidence for defining this.o   Consider moving these data into the main tables rather than in supplementary files.

o  Overall, I want more information here, particularly since these were significant findings that are then referred to in the discussion (Line 342).

We have carefully considered the information presented in the table and identified several additional methods that could be included. As a result, we have made the decision to omit this part for now. I apologize if my attempt to display a large amount of information may have caused confusion due to insufficient explanation.

However, if you believe it would be valuable to integrate this information into the main table, I would be more than happy to reanalyze the data, provide a thorough explanation, and make the necessary edits to incorporate it seamlessly. However, we consider that an additional revision period of 1 week will be required for this. Your input and feedback are highly appreciated in this process.

Discussion

  1. Line 334: Presumably another typo calling six months delayed CF introduction when this should say seven.

This study aimed to investigate the association between late CF introduction (≥ 7 months) and health outcomes in exclusively breastfed children, in comparison to introducing CF between 4-<7 months.

  1. Lines 360-361: This states that there were “no significant clinical consequences.” I would argue that low HFA and higher hospitalizations are clinical consequences. Suggest modifying this statement.
  2. R)  Thank you for the important comment. Clinical consequences are not proper based on our results, as you mentioned. We corrected the sentence as follows:

Our findings indicated that late CF introduction after 7 months, compared to introducing CF between 4-<7 months, did not result in significant childhood diseases.

  1. Lines 374-376: Please revise this sentence. Either a higher number had ICU admission or the group had an increased risk.
  2. R)  Thank you for the important comment. We corrected the sentence as follows:

Although both groups were similar in all-cause hospitalization and mortality rates, the late CF introduced group had an increased risk of ICU admission events (≥2 events), and frequent hospitalizations (≥6 events).

  1. Line 379: Based on your findings it would be more accurate to say “…CF should be introduced before seven months of age…” not “no later than.” The latter suggests that 7 months is not delayed, and this is inconsistent with your findings.
  2. R)  Thank you for the important comment. We corrected the sentence as follows:

Based on our findings, CF should be introduced before seven months of age, considering possible growth faltering in developed countries since late CF introduction may result in low HAZs.

  1. Line 386: replace the word duration for timing.

 Thank you for the important comment. We corrected the sentence as follows:

Our study indicates that late introduction of CF (≥ 7 months) in exclusively breastfed infants is linked to lower heights and increased risks for hospitalization in childhood, particularly frequent hospitalization events. However, no significant associations were found with high BMI percentiles or specific disease rates.

Conclusions

  1. Line 413: “…significant risks for hospital resources…” should be “significant risks for  hospitalization”
  2. R) Thank you for your important comment. As you commented, we revised the conclusion as follows:

Our study indicates that delaying the introduction of CF (≥ 7 months) in exclusively breastfed infants is linked to lower heights in childhood and increased risks for hospitalization, particularly frequent hospitalization events. However, it does not show an association with high BMI percentiles or specific disease rates.